# Mitochondrial DNA Changes in Genes of Respiratory Complexes III, IV and V Could Be Related to Brain Tumours in Humans

**DOI:** 10.3390/ijms232012131

**Published:** 2022-10-12

**Authors:** Paulina Kozakiewicz, Ludmiła Grzybowska-Szatkowska, Marzanna Ciesielka, Paulina Całka, Jacek Osuchowski, Paweł Szmygin, Bożena Jarosz, Brygida Ślaska

**Affiliations:** 1Department of Radiotherapy, Medical University in Lublin, Chodźki 7, 20-093 Lublin, Poland; 2Department of Radiotherapy, The Regional Oncology Centre of Lublin—St. John’s Cancer Centre, Jaczewskiego 7, 20-090 Lublin, Poland; 3Chair and Department of Forensic Medicine, Medical University in Lublin, Jaczewskiego 8b, 20-090 Lublin, Poland; 4Chair and Department of Neurosurgery and Paediatric Neurosurgery, Medical University in Lublin, Jaczewskiego 8, 20-090 Lublin, Poland; 5Institute of Biological Bases of Animal Production, University of Life Sciences in Lublin, Akademicka 13, 20-950 Lublin, Poland

**Keywords:** mitochondrial DNA, respiratory complex III, respiratory complex IV, respiratory complex V, polymorphisms, gliomas

## Abstract

Mitochondrial DNA changes can contribute to both an increased and decreased likelihood of cancer. This process is complex and not fully understood. Polymorphisms and mutations, especially those of the missense type, can affect mitochondrial functions, particularly if the conservative domain of the protein is concerned. This study aimed to identify the possible relationships between brain gliomas and the occurrence of specific mitochondrial DNA polymorphisms and mutations in respiratory complexes III, IV and V. The investigated material included blood and tumour material collected from 30 Caucasian patients diagnosed with WHO grade II, III or IV glioma. The mitochondrial genetic variants were investigated across the mitochondrial genome using next-generation sequencing (MiSeq/FGx system—Illumina). The study investigated, in silico, the effects of missense mutations on the biochemical properties, structure and functioning of the encoded protein, as well as their potential harmfulness. The A14793G (MTCYB), A15758G, (MT-CYB), A15218G (MT-CYB), G7444A (MT-CO1) polymorphisms, and the T15663C (MT-CYB) and G8959A (ATP6) mutations were assessed in silico as harmful alterations that could be involved in oncogenesis. The G8959A (E145K) ATP6 missense mutation has not been described in the literature so far. In light of these results, further research into the role of mtDNA changes in brain tumours should be conducted.

## 1. Introduction

The respiratory chain consists of four protein complexes located in the inner mitochondrial membrane and an ATP synthase called the V complex. The high efficiency of energy production is ensured by arranging carriers according to the increasing oxygen reduction potential. The first four complexes are involved in the transport of electrons to the oxygen molecule. Complex I oxidises NADH (nicotinamide adenine dinucleotide), complex II oxidises succinate to fumarate, transferring the resulting electrons to ubiquinol, which carries electrons to complex III, and then, by cytochrome c, to complex IV. Complex V uses the transmembrane proton gradient (produced by complexes I, III and IV) to produce ATP. 

Complex I is the largest component of the mitochondrial respiratory chain, and its dysfunction is the main contributor to conditions related to the dysfunction of oxidative phosphorylation [1]. One of the first discovered polymorphisms associated with cancer is a polymorphism leading to codon A114T (A—alanine, T—threonine) changes in the ND3 gene (G10398A). It was found to cause alterations in respiratory complex I [2]. Complexes I and II produce reactive oxygen species (ROS) within the mitochondrial matrix, while complex III produces ROS into either the matrix or the intermembrane space. 

Complex III includes cytochrome b, the Rieske protein and cytochrome c1 for electron transport. Complex III consists of 11 subunits, of which only cytochrome b is encoded by mitochondrial DNA [3]. Cytochrome b is part of complex III, which is the main site for proton gradient production and which mediates electron transfer from coenzyme Q to cytochrome c in the mitochondrial respiratory chain. The structure of cytochrome b includes the Qi site, which is where quinone reduction takes place. The Qi site, together with the Rieske protein, forms a hydroxyquinone oxidation site (Qo site). Cytochrome b is involved in the electron transfer between the Qi and Qo sites. It is the catalytic core of complex III and is necessary for its proper functioning. Cytochrome b comprises eight transmembrane helices and is a rather hydrophobic and conserved protein. It is believed that the *MT-CYB* encoding cytochrome b is fairly susceptible to human mitochondrial DNA changes associated with a range of conditions [4]. Complex III dysfunction may also reduce complex I activity [5]. A wide spectrum of somatic mutations in *MT-CYB* has been described in mitochondrial myopathy in the form of exercise intolerance (G15084A, G15150A, G15761A, G15723A), a condition that is partly attributable to dysfunctional energy metabolism in striated muscle cells [6]. The G15243A (G166E) and G15498A (G251D) variants are associated with hypertrophic cardiomyopathy [6,7]. Undoubtedly, the cardiac muscle and brain are sensitive to changes in mitochondrial-related oxidative metabolism. *MT-CYB* alterations have also been identified in neoplastic diseases. G15179A and A15182G alterations have been described in thyroid cancer, and T15572C has been described in colorectal cancer [8].

Cytochrome oxidase is the final component of the mitochondrial electron transport chain. Three subunits (COI, COII and COIII) are encoded by mitochondrial DNA, while the remaining ten are encoded by nuclear DNA (nDNA). Since mitochondrial subunits form the functional core of cytochrome oxidase, their proper functioning may clearly play a significant role in oxidative phosphorylation. Although complex IV is believed to be the smallest contributor to reactive oxygen species production, it has been noted that changes in nDNA genes encoding cytochrome oxidase subunits may lead to increased ROS production through other signalling pathways, such as RAS. Cytochrome oxidase dysfunction has been reported in various conditions. For instance, in people with Alzheimer’s disease, the cytochrome oxidase complex has been found to exhibit low activity levels [9]. COI is the main catalytic subunit of the cytochrome oxidase complex. It collects electrons from COII to its active sites—two heme systems and the binuclear copper centre. Changes in COI reduce cytochrome oxidase complex activity, leading to increased production of ROS and nitric oxide, and may further cause increased tumour proliferation [1,3]. Increased amounts of SNPs (single nucleotide polymorphisms) in *MT-CO1* were found in prostate cancer [10]. Lueth et al. identified 34 mtDNA somatic mutations in 84% (16 out of 19) of patients with pilocytic astrocytoma. Of these, 17 were found in genes of oxidative phosphorylation. Three of them were missense mutations—CO1 (L112 M), cytochrome b (L236I) and ATP6 (M60 V) [11].

Reduced COI activity was noted in colon cancer cells, which correlated with a resistance to apoptosis and a higher level of free radicals [8,12]. Four non-synonymous *MT-CO1* polymorphisms in prostate cancer cells, related to specific mitochondrial haplogroups, have been found [13]. The T6253C (M117T) polymorphism was identified in cancer cells of three individuals belonging to the H haplogroup, and the G6261A (A120T) polymorphism was found in six patients from four different haplogroups (J, T, L1 and N). Furthermore, the C6340T (T146I) transition was found in one patient in the H haplogroup and one patient in the N haplogroup, while the A6663G (I254V) transition was identified in five patients from two different haplogroups (O and L2). [10]. The incidence of COI gene polymorphisms in the American population of African and European origin was 17.4% and 6.5%, respectively. This indicates a higher risk of prostate cancer in the African group [14]. The T7389C (Y496H) polymorphism and the synonymous T6221C polymorphism were significantly correlated with prostate cancer (*p* < 0.05) [14]. In their study of the American population (482 men with prostate cancer and 189 men without cancer), Scott et al. found *MT-CO1* polymorphisms in 8.8% of Caucasian patients with prostate cancer and 72.8% of patients of African descent. In the control group, the percentage of polymorphisms in Caucasian men was 0.0%, as opposed to 64.3% in African Americans [15].

ATP synthase is involved in transporting the proton gradient across the mitochondrial membrane for ATP synthesis. Apart from its clear role in oxidative phosphorylation, it is also involved in the correct apoptosis [16]. Changes in the genes encoding the ATP6 and ATP8 synthase subunits may disrupt proton transport through the mitochondrial membrane, reduce ATP production and contribute to an increase in mitochondrial membrane potential [16,17]. One study of osteosarcoma found 23 *ATP6* mutations in 24 out of 39 patients, as well as four missense polymorphisms [18]. *ATP6* and *ATP8* alterations have also been described in bladder cancer and epithelial ovarian tumours [19]. Mitochondrial haplogroups have been shown to play a role in neurological diseases. The risk of Parkinson’s disease is higher in people with haplogroups J and T, characterised by T to C substitution at position 4216 of the ND1 gene [20,21]. As regards neurodegenerative conditions, one study involving a European population has shown an increase in the risk of Alzheimer’s disease among males from haplogroup U, compared to haplogroup H, the most common haplogroup in Europe [22].

The aim of this study was to identify the possible relationships between brain gliomas in humans and the occurrence of specific mitochondrial DNA polimorphisms and mutations in respiratory complexes III, IV and V.

## 2. Results

### 2.1. Polymorphisms in Complex III Genes

The results for polymorphisms in complex III genes (cytochrome *bc1*) are presented in Appendix A. Appendix A provide an assessment of the effects of amino acid-changing polymorphisms on biochemical properties and of their potential to affect protein function. 

In the investigated material, a total of 19 different polymorphisms were detected in 30 patients, of which nine were missense and 10 were synonymous. Two polymorphisms (C15499T and A15656G) have not been described in the literature so far. In the case of the synonymous A15244G polymorphism, heteroplasmy occurred in both blood and tumour samples (Appendix A). This change was found in only one patient diagnosed with grade IV glioma (Appendix A). 

The most common polymorphism was the A15326G (T194A) missense variant, identified in all 30 patients (Appendix A). The incidence of this change in the mtDB (Human Mitochondrial Genome Database) was presented as high (2687) relative to the reference sequence (17) and concerned the variable region (Appendix A). The variant caused a decrease in the grand average of hydropathicity, from 0.693 to 0.637, and a more than two-fold increase in the percentage of the twelfth helix, from 2.00% to 4.59% (Appendix A). In the PSSM assessment, threonine scored 2 points and alanine scored 3 points (Appendix A). 

The second most common missense polymorphism was T14766C (I7T)—36.6 (6)% (11 out of 30) (Appendix A). According to the mtDB, the change was fairly common (610) and concerned the variable region (Appendix A). It caused a decrease in the grand average of hydropathicity and a slight increase in the aliphatic index of the protein (Appendix A). In the PSSM assessment, isoleucine at position 7 scored −2 points and threonine scored 5 points (Appendix A). 

The change in the T14798C (F18L) missense polymorphism was found at a rate of 6.6 (6)% (2 out of 30) (Appendix A). This change was in the variable region (3 points) (Appendix A) and caused a decrease in the grand average of hydropathicity of the protein, with no significant deviations in other protein properties (Appendix A).

The C15452A (L236I) missense polymorphism caused a decrease in the grand average of hydropathicity of the protein, from 0.693 to 0.633, and in the percentage of the fourteenth helix, from 14.50% to 13.36% (Appendix A). It occurred at a rate of 10% (3 out of 30) (Appendix A). In the PSSM assessment, leucine at position 236 scored 4 points, the same as isoleucine (Appendix A). This variant occurred in the variable region (Appendix A).

The A15218G (T158A) missense polymorphism caused a decrease in the grand average of hydropathicity, from 0.693 to 0.637, and a slight increase in the stability index, from 41.02 to 42.08 (Appendix A). It was identified in two patients diagnosed with grade IV glioma and in one patient with grade II glioma (Appendix A).

Only the A14793G (H16R) missense polymorphism caused a change in the isoelectric point, as it increased from 7.83 to 8.42, without changing the aliphatic index, but with a slight increase in the stability index and a decrease in the grand average of hydropathicity (Appendix A). In the PSSM assessment, histidine at position 16 scored 8 points and arginine scored 0 points (Appendix A). 

The C15459T (S238F) missense alteration occurred in only one of the studied patients diagnosed with glioblastoma (Appendix A). The incidence of this variant in the mtDB proved to be very low, with only one case identified, which, however, did not concern the conserved region (2 points) (Appendix A). The change caused a decrease in the grand average of hydropathicity and a significant increase in the percentage of the fourteenth helix, from 14.50% to 32.29% (Appendix A). In the PSSM assessment, both serine and phenylalanine scored 0 points (Appendix A). 

The A15656G (I304V) missense polymorphism was not described in the mtDB. It occurred in one patient diagnosed with stage IV glioma (Appendix A). The change concerned a conserved amino acid (5 points) (Appendix A). It slightly decreased the aliphatic index and the average hydrophobicity of the protein. However, this polymorphism did not affect the isoelectric point (Appendix A). In the PSSM assessment, at position 304, isoleucine scored 6 points and valine scored 2 points (Appendix A). 

The last of the missense polymorphisms, A15758G (I338V), caused a decrease in the grand average of hydropathicity of the protein without significant deviations in other biochemical properties of the protein (Appendix A). It was found in one patient diagnosed with glioblastoma (Appendix A). In the PSSM assessment, isoleucine at position 338 scored 7 points and valine scored 2 points (Appendix A). The change occurred in a moderately conserved region, which scored 4 points (Appendix A).

As assessed in the SIFT Sequence program, four missense polymorphisms may have influenced protein function (A14793G, A15218G, C15452A and A15758G), while the remaining five probably did not have this effect (Appendix A). In the APOGEE assessment, only the A15758G (I338V) variant was pathogenic, while the T14766C (I7T) variant reached the cut-off value of 0.5 (Appendix A). The etched missense polymorphisms did not cause any helix deviation from the correct sequence defined by the TMHMM Server. The most common synonymous polymorphism was G14905A, which occurred at a rate of 6.6 (6)% (2 out of 30) (Appendix A).

### 2.2. Polymorphisms in Complex IV (Cytochrome Oxidase) Genes

The results for polymorphisms in complex IV (cytochrome oxidase) genes are presented in Appendix A. Appendix A present an assessment of the effects of amino acid-changing polymorphisms on biochemical properties and of their potential to affect protein function. 

A total of 21 types of polymorphisms were detected in cytochrome oxidase complex genes, with 39 cases found overall in the studied patients. In *MT-CO1*, out of the 13 different polymorphisms, two were missense alterations (G7444A and G7075C). A total of five types of polymorphisms were detected in *MT-CO3*, one of which was missense (G9477A) and four were synonymous. Only three synonymous polymorphisms appeared in *MT-CO2*. The most common cytochrome oxidase complex polymorphism was the synonymous C7028T change in COI, which occurred at a rate of 43.3 (3)% (13 out of 30) (Appendix A). The G9477A (V91I) missense polymorphism found in COX3 was identified in 10% (3 out of 30) of the studied patients, including one patient in each of the three grades (II, III and IV) (Appendix A). The change did not cause any deviation in the protein isoelectric point or in the alpha percentage of the third helix. However, it led to a slight increase in the stability index, from 22.96 to 23.70, and a minimal increase in the aliphatic point and the grand average of the hydropathicity of the protein (Appendix A). In the PSSM assessment, valine at position 91 scored 6 points and isoleucine scored 3 points (Appendix A). The normalised conservativeness score at V91 was −0.298, indicating a conserved region (6 points) (Appendix A).

The missense change that occurred in COI (G7444A) resulted in a change in the AGA stop codon reading and was found in only one patient diagnosed with grade IV glioma (Appendix A). The change increased the isoelectric point from 6.19 to 6.29 while slightly decreasing the aliphatic index, the stability index and the grand average of hydropathicity of the protein (Appendix A). The G7075C (G391A) missense polymorphism did not cause any significant changes in the protein’s biochemical properties, apart from an almost two-fold percentage increase in the nineteenth helix, from 0.25% to 0.47% (Appendix A). It was found in one patient with a stage IV tumour diagnosis (Appendix A). In the PSSM assessment, at position 391, glycine scored 2 points and alanine scored 5 points (Appendix A). The change concerned the conserved region (6 points) (Appendix A). For missense changes of the cytochrome oxidase complex, no helical deviations from the correct sequence were found using the TMHMM Server other than the extension of the amino acid sequence for the G7444A alteration in the mitochondrial matrix (Appendix A).

### 2.3. Polymorphisms in Complex V Genes

The polymorphism results for ATP synthase subunit 6 genes are presented in Appendix A. Appendix A present an assessment of the effects of amino acid-changing polymorphisms on biochemical properties and of their potential to affect protein function. No polymorphisms were detected in the ATP synthase subunit 8. In *MT-ATP6*, a total of nine different polymorphisms were detected in 30 patients, including two missense (A8860G and G9055A) and seven synonymous polymorphisms. The most common change was the A8860G (T112A) sense polymorphism, which occurred in all patients (30 out of 30) (Appendix A). The rate of this change in the mtDB was assessed as high (2698) relative to the reference sequence (6) (Appendix A). The normalised conservativeness score in the T112 position was −0.463, indicating a conserved region (Appendix A). The change slightly increased the percentage of the ninth helix, from 0.07 to 0.12 (Appendix A). In the PSSM assessment, threonine at position 112 scored 6 points and alanine scored 2 points (Appendix A).

The G9055A (A177T) sense polymorphism, found in patients with grade II and III gliomas, decreased the percentage of the fifth helix from 1.18% to 0.77% and slightly reduced the grand average of hydropathicity from 0.901 to 0.890 (Appendix A). Neither of the missense polymorphisms caused any deviations in the isoelectric point or the protein stability index, with only slight changes in the aliphatic index (Appendix A). The G9055A (A177T) polymorphism occurred in 10% (3 out of 30) of the patients (Appendix A). In the PSSM assessment, alanine at position 177 scored 6 points and threonine scored 1 point (Appendix A). Alanine at position 177 was shown to be an amino acid in the conserved region (Appendix A). As analysed in SIFT Sequence, both missense polymorphisms were assessed as harmless to protein function (Appendix A) and did not cause any helix deviation from the correct sequence defined by TMHMM Server. The pathogenicity predictor, APOGEE, assessed them as neutral changes (Appendix A).

### 2.4. Mutations

The detected mutations are provided in Appendix A. The *CYB* region mutation proved to be the T15663C (I306T) (Appendix A) missense polymorphism, causing a slight decrease in stability and aliphatic indices, and in the average hydrophobicity. The protein isoelectric point did not change (Appendix A). This mutation occurred in one patient diagnosed with grade IV glioma (Appendix A). A synonymous G6755A mutation in *COI* was detected in a single patient diagnosed with stage IV glioma. One G8959A (E145K) missense mutation was detected in the *ATP6* region, which increased the isoelectric point from 10.09 to 10.35 and decreased the protein stability index from 35.26 to 32.97, leaving the protein stable (Appendix A). In the PSSM assessment, glutamic acid at position 145 scored 7 points and was the preferred amino acid in that position (Appendix A). The assessment of the conservativeness of the missense mutations and of their potential to affect protein function is presented in Appendix A.

## 3. Discussion

In the investigated material, *MT-CYB* missense polymorphisms were relatively common (9 out of 19), the most common among all mitochondrial subunits (9 out of 34). Moreover, numerous synonymous polymorphisms occur in this region. Therefore, cytochrome b alterations may be involved in the formation of brain tumours. *MT-CYB* changes were found to be highly representative of glioblastoma [4]. Rhiannon et al. (2015) [4] conducted a study of mitochondrial DNA complexes III and IV in 32 glioblastomas, investigating tumour and blood samples. Several polymorphisms were detected that partially overlapped with the variants detected in the present study. Most of the changes concerned *MT-CYB*. The C14766T (T7I) polymorphism was present in 14% of glioblastomas, while in the present study, it was found in 37% of cases. One of the more common polymorphisms in the cited study, as well as in the present study, was the A15326G (T194A) polymorphism (23.81% and 100%, respectively). This polymorphism is not specific to any mitochondrial subgroup. Other *CYB* polymorphisms identified in both the present study and other studies included T14798C (F18L), A14793G (H16R), C15452A (L236I) and A15758G (I338V) [4]. The authors of the cited studies concluded that the T14798C (F18L) transition may have interfered with ubiquinone binding to complex III, thus potentially affecting protein function. This transition was found in 30.95% of the patients [4]. However, an assessment using the ConSurf Database suggested that the mutation did not concern the conserved region (3 points), while SIFT Sequence classified it as harmless to protein function.

In another study, this variant was present in glioblastoma in 11 out of 32 patients, as well as in ovarian serum adenocarcinoma (6 out of 28) and colon adenocarcinoma (11 out of 86) [19]. In pituitary adenoma, this variant occurred as a somatic mutation, which indicates its pathogenicity [3,5]. In the present study, the T14798C (F18L) transition as a polymorphism was found in two patients, including one patient with a stage II tumour and one patient with a stage III glioma. This change is specific to Asian mitochondrial subgroups, such as Q1f1 and A2ad1, as well as European subgroups: J1c, T2g and K. F18L. In the Uniprot database, it was recorded as a natural cytochrome b variant [23]. The similarities between the findings of this study and the results of the studies cited above suggest that the described mutations may be related to brain neoplasms.

Also of note is the A15758G (I338V) polymorphism, which was assessed as potentially affecting protein function. Such polymorphism has already been described in studies involving glioblastoma [4]. Initially considered harmless, it was subsequently shown as pathogenic by the APOGEE predictor. There are also reports of the A15758G (I338V) variant as a somatic mutation in thyroid and prostate cancer, and in endometrial cancer of the uterus [7,8]. The harmfulness of this mutation is also indicated by the fact that at position 338, isoleucine was preferred over valine. This could suggest a favourable change for the protein. Polymorphism was found in one patient diagnosed with stage IV glioma, so it is difficult to draw conclusions about its involvement in tumour formation. The mutation belongs to numerous subgroups of European (I, H, J, T, U, K), Asian (G, D, B) and African (L3) haplogroups.

The C15452A (L236I) polymorphism was described in 30.95% of the patients with glioblastoma [4] and as a somatic mutation in patients with pituitary adenoma and low grade pilocytic astrocytoma [5,11]. Another study found it not only in glioblastoma (11 out of 32) but also in colon adenocarcinoma (18 out of 86), acute myeloid leukemia (7 out of 37), ovarian serum adenocarcinoma (5 out of 28) and rectal adenocarcinoma (14 out of 43) [24]. It was assessed as potentially affecting protein function, although leucine and isoleucine were equally preferred at position 236, and the change concerned the variable region. The variant is specific to the European JT haplogroup and has been reported as common in patients with bladder cancer [25]. It is difficult to make conclusive judgments about the harmfulness of this change, especially since the Uniprot database recorded the L236I as a natural cytochrome b variant [23]. However, its presence in many types of cancer, including brain gliomas, suggests that it may play a role in oncogenesis.

The A15218G (T158A) polymorphism, similarly to A15758G and C15452A, has been described in glioblastoma and assessed as potentially affecting protein function [4]. What is more, in one study, the A15218G (T158A) polymorphism was present in 2% of women at an increased risk of breast cancer without *BRCA 1* and *BRCA 2* mutations [26]. It has also been reported in colon adenocarcinoma (4 out of 86), rectal adenocarcinoma (2 out of 43), glioblastoma (1 out of 32) and acute myeloid leukemia (1 out of 37) [18,19]. SIFT Sequence assessed it as a harmful change, and this was further corroborated by PSSM. The change concerned the variable region, but it is relatively rare (38) compared to the reference sequence (2665). In the present study, it was found in three patients, including one patient with stage II glioma and two patients with stage IV glioma. The polymorphism is specific to the Asian (M7a1a2 and M10a1) and European (HV1a′b′c, H13a2c and U5a1) mitochondrial subgroups.

The last of the potentially harmful polymorphisms found in the material tested is A14793G (H16R). This change has already been reported in brain glioblastomas [4]. It was also identified as a polymorphism in colon adenocarcinoma (6 out of 86) or acute myeloid leukemia (3 out of 37), and as a somatic mutation in prostate cancer and type I endometrial cancer [24]. Like T158A, the H16R variant concerned the highly variable region. It increased the isoelectric point in silico from 7.83 to 8.42. This variant is relatively rare (39) compared to the reference sequence (2665). In the present study, it was found in three patients. Given the literature data and the in silico assessment, the relationship between oncogenesis and both the A15218G (T158A) and A14793G (H16R) polymorphisms cannot be ruled out.

Other missense polymorphisms detected in *MT-CYB* (A15326G, C15459T and T14766C) had no effect on protein function when assessed in silico. The C15459T (S238F) polymorphism is specific not only to the Asian mitochondrial subgroups (F1b1d and B4b1a2d), but also to the African L0d1d and the European H7c3 groups. It affects the transmembrane section of the protein in a region of low complexity. Apart from the T14766C transition described in breast cancer [25], there are no reports on the relationship between the above polymorphisms and any specific conditions. 

In the material investigated, the only missense mutation was detected in *CYB* (T15663C (I306T)), and it concerned the conserved region (5 points). Threonine was less preferred at this protein position than isoleucine. This suggests that the mutation is harmful despite its positive assessment in the SIFT Sequence. This change does not appear in the database as a polymorphism specific to European subgroups, but it is specific to one of the mitochondrial subgroups within the African L1 haplogroup, as well as the Asian M and S groups. The mutation has been described in cataract patients [27]. In one of the studies where the influence of this mutation on complex III function was assessed in silico, its potential harmfulness was confirmed. Above all, it was shown to interfere with the function of the heme involved in the transport of electrons in the respiratory chain. This conclusion was made thanks to the complex III models developed in silico, in which the location of the I306T mutation indicated a possible interaction with the b heme and, additionally, a reduction in the stability of complex III [27]. In conclusion, due to the importance of cytochrome b in the respiratory chain, mutations in *MT-CYB* may contribute to brain tumour formation.

In the investigated material, most of the changes among the three mitochondrial subunits of complex IV concerned *MT-CO1*, although this region is considered to be highly conserved. Two missense polymorphisms (G7444A and G7075C) were detected in the region in two different patients with stage IV tumours. The *MT-CO1* G7075C (G391A) transition has not been described in the literature so far and was not found in the available mtDNA databases. The Sift Sequence classified the variant as harmless. The APOGEE predictor assessed the G391A change as neutral, although with a borderline score of 0.49.

The G7444A (X514K) polymorphism changed the reading in the stop codon (AGA) number 514, resulting in an extension of the *COI* polypeptide by three amino acids (Lys–Gln–Lys) to the C-terminus of the polypeptide. Interestingly, this variant is adjacent to the site of the endonucleolytic processing of the 3’ end of the L–strand RNA precursor, which includes tRNASer (UCN). Hence, this change may affect the functioning of this subunit and, consequently, of the entire mitochondrion. No changes in the adjacent tRNASer sequences were found in the patient with the detected polymorphism. A change in the protein isoelectric point may affect protein function. Moreover, the change is rare in the population (10) compared to the reference sequence (2694). Co-occurring with C1494T (12S rRNA), the G7444A alteration was described by Chinese researchers as increasing the risk of aminoglycoside-induced non-syndromic hearing loss [28]. In the investigated material, no coexistence with the C1494T change was found. The G7444A polymorphism was also identified by Scott et al. in patients with prostate cancer (0.2% of the patients) [15]. The G7444A variant appeared in mitochondrial subgroups found in Europe (W4b, V7, H40b, R8a1a1a2) and Asia (D4a6).

COIII appears to be involved in proton transport as an integral part of the mitochondrial membrane. Changes in the gene that encodes this subunit have not been reported as often as those in *COI* [29]. The *MT-CO3* G9477A (V91I) missense polymorphism detected in the investigated material was described in serum ovarian cancer and was found to increase the risk of this cancer [30]. In studies on glioblastoma-related mtDNA changes in mitochondrial complex IV, most variants were found in *MT-CO1* and *MT-CO3* [4]. However, they did not find any changes in *MT-CO2.* In the investigated material, only synonymous polymorphisms were detected in *MT-CO2* (T8038G, G8251A and G8269A). The G9477A polymorphism was found in 4.76% of glioblastomas [4]. Polymorphism is rare (98) and specific to the U5 mitochondrial subgroup. In the investigated material, this polymorphism was found in three cases, and in each patient, it was accompanied by missense changes in *MT-CYB* (C14766T, A14793G, A15218G and A15326G). In the Uniprot database, V91I has been presented as a naturally occurring variant in COIII [29]. Although polymorphism itself does not appear to be detrimental to mitochondrial function, its coexistence with *MT-CYB* missense polymorphisms, found in three patients with varying degrees of tumour differentiation, may suggest its involvement in oncogenesis. It should also be noted that all the three patients belonged to the U5a haplogroup, which included three out of thirty patients in the investigated group (unpublished data).

In one study on the PC3 prostate cancer cell line, the pathogenicity of changes in mitochondrial DNA was confirmed precisely based on mutations in *ATP6* (T8993G) [13]. It changed leucine to arginine at protein position 156 (L156R), resulting in a 70% decrease in ATP6 synthase activity and a significant increase in the production of free radicals. After implanting cells with this transition in mice, increased proliferation of these cells was noted, suggesting that this transition could be involved in tumour progression [13]. Ghaffarpour et al. (2014), examining tumour samples collected from 49 Iranian patients with breast cancer, found 28 changes in complex V mitochondrial genes that were not present in healthy tissue. Of these 28 variants, 23 were related to *ATP6* [31]. It can be assumed that the ATP6 encoding gene is more susceptible to changes associated with oncogenesis. Among the changes found by the above-mentioned study, two missense variants coincided with those found by the present study (G8860A and G9055A) [31]. The 8860G polymorphism is more frequent (2698) compared to the reference variant 8860A (6). Accordingly, in the present study, the A8860G change occurred in all investigated tumours (30 out of 30). There are studies suggesting that this polymorphism is more common in patients with breast cancer [31,32,33]. In the study by Aikhionbare et al., polymorphism was present in 96 out of 102 investigated ovarian tumours [34]. In an Iranian study, this change was identified in 30 out of 31 patients with hypertrophic cardiomyopathy [35]. In another study, it was found in all 23 studied patients with bipolar disorder [36], and in yet another, in all patients with Alzheimer’s disease [37]. Since the variant involved a conserved amino acid (6 points) and appeared in so many conditions, the G8860A polymorphism may be expected to affect the functioning of the mitochondrial respiratory chain, even though SIFT Sequence classified it as harmless. It is possible that this variant adapts the cells to a changing microenvironment and helps them survive in new conditions.

The G9055A (A177T) polymorphism belongs to the subgroups representing the European haplogroups (R0b, H5a, J1c, U5a and U8b), and also the Asian ones (M37a, M71c, M76a, Z3b, A2x, A2a and B4b) and is not as common (134) as the reference 9055G (2570). There are studies describing a relationship between the higher risk of breast cancer progression and the occurrence of the G9055A polymorphism [2,38]. These variations have also been reported in pancreatic cancer [39] and non-invasive bladder cancer [40]. One study showed a reduced risk of Parkinson’s disease in Caucasian women with such a polymorphism [21]. This mutation involved a conserved amino acid (6 points) and while considered harmless by SIFT Sequence, its impact on the functioning of mitochondria cannot be ruled out. In the Uniprot database, the A177T and T112A transitions were found as natural *ATP6* variants [41].

One G8959A (E145K) missense mutation was also found in the *ATP6* region—in one patient diagnosed with stage III glioma. Studies on *Saccharomyces cerevisiae* have shown that mutations in the P136S and K64E protein positions, found in prostate and thyroid cancer, increase the susceptibility of yeast cells to compounds causing oxidative stress. The 8932C > T (P163S) mutation decreased the effectiveness of the ATP synthase complex [41]. The G8959A (E145K) mutation involved a highly conserved amino acid (9 out of 10 points) and was assessed as harmful in the SIFT Sequence. The mutation does not appear in the PhyloTree.org or mtDB databases, and there are no reports of any specific conditions associated with it. The change may play a particular role in the initial stages of oncogenesis by increasing free radicals. One study involving sideroblastic anaemia found a mutation located close to the one in question—in the S148N codon [42]. Undoubtedly, its effects on the functioning of the mitochondrial respiratory chain and its potential relationship with oncogenesis should be taken into account. 

### Summary

Three *MT-CYB* polymorphisms: A14793G, A15758G and A15218G, as well as the *MT-CO1* G7444A, *MT-CYB* T15663C and G8959A (ATP6) mutations, may affect the function of mitochondria, thus potentially playing an important role in the formation of brain tumours. The T15663C (*MT-CYB*) mutation may be a predisposing factor in brain tumours for the Caucasian population since it belongs only to the mitochondrial subgroups of haplogroups (M, S, L1) outside this population. In light of these results, further research into the role of mtDNA polymorphisms and mutations in brain tumours should be conducted.

## 4. Materials and Methods

### 4.1. Material

The investigated material consisted of blood samples and tumour tissue fragments collected during brain tumour removal surgeries performed between 2016 and 2019 at the Department of Neurosurgery and Pediatric Neurosurgery of Independent Public Clinical Hospital No. 4 in Lublin. The study group comprised 30 Caucasian patients. The investigated material was collected from patients diagnosed with grade II, III or IV glioma. As the new WHO classification introduced in 2016 includes the tumour genotype, the study additionally determined the mutation status of *IDH* (isocitrate dehydrogenase) genes in tumour tissue. Changes affecting codon 132 of the *IDH1* gene and codon 172 of the *IDH2* gene were examined using Sanger sequencing. The patients with brain tumours had received no previous oncological treatment. They ranged in age from 22 to 76, with a mean age of 50.2. The patient profile is presented in Appendix A**.**

### 4.2. Methods

DNA was isolated from brain tumour samples and peripheral blood using a DNA isolation kit (DNeasy Blood & Tissue Kit, Qiagen, Hilden, Germany) according to the manufacturer’s instructions. The concentration and purity of the obtained DNA were evaluated using a NanoDrop 1000 spectrophotometer (Thermo Scientific, Waltham, MA, USA). The sequencing procedure was performed using the Nextera XT DNA Library Preparation Kit and the Nextera XT Index Kit (Illumina), with the following sequencing steps: amplification of the entire mtDNA genome, purification of the obtained products and the assessment of their concentration, preparation of libraries and high-throughput sequencing performed using the MiSeq device (Illumina) with the chemistry-based kit MiSeq Reagent Kit v.3 (600 cycles). The integrated sequencer software—MiSeq Control Software, Real-Time Analysis Software, MiSeq Reporter—was employed to run and control the process, and to generate files in the FASTQ format.

The sequenced genomes were compared with the reference sequence of the human mtDNA (GenBank NC_012920, AC_000021) using the mtDNA Variant Processor and mtDNA Variant Analyzer software available on the Base Space Illumina platform. The changes obtained from the neoplastic tissue were compared with variants from the blood of the respective patients. Any mtDNA alteration occurring in both blood and tumour tissue was identified as a polymorphism (a germline change). An mtDNA alteration occurring only in tumour cells was identified as a mutation (a somatic change). The affiliation of mtDNA polymorphisms to a given mitochondrial subgroup was established using PhyloTree.org—mtDNA tree Build 17 function available on https://www.mitomap.org/MITOMAP. The detected missense polymorphisms and mutations were analysed in silico using the following bioinformatics software:

The detected missense polymorphisms and mutations were analysed in silico using the following bioinformatics software:ExPASy Proteomics tools—ProtParam Program—assessment of protein parameters, such as the heoretical isoelectric point (pI), protein stability index, aliphatic index and grand average of hydropathicity [https://www.expasy.org/resources/protparam; accesed on 9 November 2019] [43].Pfam version 33.1—assessment of the secondary protein structure, including the number of helices [http://pfam.xfam.org/search#tabview=tab1; accesed on 29 September 2020] [44].AGADIR—theoretical prediction of the helix percentage in a given segment of the protein [http://agadir.crg.es/protected/academic/calculation4.jsp; accessed 11 September 2020] [45].TMHMM 2.0—assessment of the presence of transmembrane segments in a given protein with a specific sequence of amino acids [http://www.cbs.dtu.dk/services/TMHMM/; accessed on 4 July 2020] [46].PSSM Viewer—assessment of the incidence of amino acids in each alignment of a protein sequence [http://www.ncbi.nlm.nih.gov/Class/Structure/pssm/ pssm_viewer; acccesed 20 October 2020] [47].SIFT Sequence—assessment of the effects of changing the sequence of amino acid residues on protein function. Score: tolerated change > 0.05, change affecting protein function ≤ 0.05 Score: tolerated change > 0.05, change affecting protein function ≤ 0.05 [https://sift.bii.a-star.edu.sg/www/SIFT_seq_submit2.html; accessed on 2 November 2020] [48].MitImpact 3D-predictor APOGGE—mitochondrial database with a predictor assessing the potential pathogenicity of changes. Score: neutral change ≤ 0.5, pathogenic change > 0.5 [https://mitimpact.css-mendel.it/; accessed on 15 October 2020] [49].ConSurf-Database—analysis of amino acid conservation in a given protein alignment. Score on a 1–9 scale (1–3 variable region, 4–6 moderately conserved region, 7–9 highly conserved region) and normalised points (variable region < 0, moderately conserved region 0–0.5, highly conserved region > 0.5) [https://consurfdb.tau.ac.il/index.php; accesed on 9 October 2020] [50,51].

## Data Availability

All data are included in the article and Appendix A.

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
