# Peer review of "Mitochondrial DNA Changes in Genes of Respiratory Complexes III, IV and V Could Be Related to Brain Tumours in Humans"

_ijms, 2022, doi:10.3390/ijms232012131_

Round 1
Reviewer 1 Report
Major comments:
The study of Kozakiewicz et al. addresses the role of changes in mtDNA genes in brain tumor. Whereas it is a very interesting topics, and the results presented here can help to understand it, the way it is currently presented does not allow the reader to follow the idea and to capt the interest of the results.
Specific comments:
Abstract
· The abstract needs complete reorganization in order to get the attention and the understanding of the reader.
· Line 17: The first sentence is not clear: ‘ …. in the production of the necessary ATP carrier…’ should the word carrier be removed?
· Line 18-19: There is no link between the two sentences. One sentence is missing? It goes from very general information on the mitochondrial respiratory system, and specific material used for the study.
· COME BACK TO THE ABSTRACT AFTER LOOKING AT THE REST.
Introduction
· The organization of the introduction needs to be completely reviewed. The way it is presented, it is very hard to read and to follow the general idea.
· Line 34: The abbreviation for ATP does not need to be defined.
· Line 35 is repetitive from the abstract, it is probably fitting better in the introduction.
· Line 37: It is not clear why they start with talking about complex III, and do not address complexes I and II.
· Lines 39-40: The separation between the paragraphs seems to be not at the appropriate place? I think the first paragraph should mention why Complex III, IV and ATP synthase are the parts of the respiratory system on which they emphasis, and then, the specific aspect on Complex III can be in the next paragraph.
· Because complex III is the hardest to measure activity, there is less data on this complex than any other. So I’m not sure the arguments presented here to justify the emphasis on complex III are appropriate and convincing. Complex I changes have been well studied and shown in many disease entities, and is not covered here, why?
· Lines 47-60: It is here a mix of data on the importance of Complex III. Maybe the focus should be to the nervous system as the paper is on brain tumor. The changes could be completely different in breast cancer. Even in various types of tumor, it varies.
· Lines 64-100: It also review the importance of Complex IV in various tissues and diseases. It should be more specific to aspect that are link to the project presented here.
· Lines 101-102: The paragraph on ATP synthase is unclear. Different aspect are presented vaguely and it is hard to make a clear link with the current study.
Results
· The results should be organized more clearly. They are presented more like a list of results. The paper includes too many tables, with list of results, which should be part of supplementary data.
· It is very difficult to follow the idea with the results presented this way.
Discussion
· It is not clear what the study presented here brings in addition of the results presented in Rhiannon et al. (2015) and other discussed here.
· The discussion is way too long, difficult to follow, and does not clearly show what the current study brings to the understanding of the role of mitochondria in brain tumor.
· A conclusion is missing to finish the paper.
Methods
· The table 1 is cited at the end of the paper. The figures should be numbered in order of citation.
Author Response
We would like to thank the reviewers for their generous comments on the manuscript. We have edited the manuscript to address their concerns. Point-by-point responses to reviewers’ comments are listed below this letter
Response to Reviewer 1:
Abstract
- The abstract needs complete reorganization in order to get the attention and the understanding of the reader. – the abstract was corrected
- Line 17: The first sentence is not clear: ‘ …. in the production of the necessary ATP carrier…’ should the word carrier be removed? –The abstract was changed
- Line 18-19: There is no link between the two sentences. One sentence is missing? It goes from very general information on the mitochondrial respiratory system, and specific material used for the study. – the abstract was changed
Introduction
- The organization of the introduction needs to be completely reviewed. The way it is presented, it is very hard to read and to follow the general idea.
- Line 34: The abbreviation for ATP does not need to be defined. – It was corrected
- Line 35 is repetitive from the abstract, it is probably fitting better in the introduction. – Abstract was corrected
- Line 37: It is not clear why they start with talking about complex III, and do not address complexes I and II. – In the introduction, we focused on the complexes studied by us, so the other complexes have not been described. The following sentences have been added:
The first four complexes are involved in the transport of electrons to the oxygen molecule. Complex I oxidizing NADH (nicotinamide adenine dinucleotide), complex II oxidizing succinate to fumarate , transferring the resulting electrons to ubiquinol, which carries electrons to the complex III, and then by cytochrome c to the complex IV. Complex V uses the transmembrane proton gradient (produced by Complex I, III and IV) to generates ATP. “
Lines 39-40: The separation between the paragraphs seems to be not at the appropriate place? I think the first paragraph should mention why Complex III, IV and ATP synthase are the parts of the respiratory system on which they emphasis, and then, the specific aspect on Complex III can be in the next paragraph.
Because complex III is the hardest to measure activity, there is less data on this complex than any other. So I’m not sure the arguments presented here to justify the emphasis on complex III are appropriate and convincing. Complex I changes have been well studied and shown in many disease entities, and is not covered here, why? -???
We studied changes in complex III, IV and ATP synthetase. The aim of the study was not to study mutations in complex I. Therefore, complex I was not taken into account. Detailed discussion of this complex would extend the already long introduction. Taking into account the reviewer's suggestion, we have added the following content
The first four complexes are involved in the transport of electrons to the oxygen molecule. Complex I oxidizing NADH (nicotinamide adenine dinucleotide), complex II oxidizing succinate to fumarate , transferring the resulting electrons to ubiquinol, which carries electrons to the complex III, and then by cytochrome c to the complex IV. Complex V uses the transmembrane proton gradient (produced by Complex I, III and IV) to generates ATP. Complex I is the largest component of the mitochondrial respiratory chain and its disruption has the greatest impact on the development of diseases related to the dysfunction of oxidative phosphorylation [1]. One of the first discovered polymorphisms associated with cancer was in complex I leading to a change in codon A114T (A – alanine, T – threonine) in the ND3 gene – G10398A. It caused alterations in the respiratory complex I [2] Complex I and II generate reactive oxygen species (ROS) within the mitochondrial matrix while complex III generates ROS into either the matrix or the intermembrane space.
Lines 47-60: It is here a mix of data on the importance of Complex III. Maybe the focus should be to the nervous system as the paper is on brain tumor. The changes could be completely different in breast cancer. Even in various types of tumor, it varies.-
We wanted to show that mutations in cytochom b are describe in cancers that which may indicate a relationship between the changes and the carcinogenesis process, especially if these changes are repeated in various cancers. As suggested by the reviewer, the following sentences were removed:
Most reports concern breast cancer, such as the missense polymorphisms that appear among Senegalese women: A15548G, A15564C, A15535G, or the following mutations: C15664CA, A15689T, T15787G [2]. In the MITOMAP database, there are variants in MT-CYB that have been described in endometrial and ovarian cancer, such as the somatic mutation T15573C.
- Lines 64-100: It also review the importance of Complex IV in various tissues and diseases. It should be more specific to aspect that are link to the project presented here.
- Lines 101-102: The paragraph on ATP synthase is unclear. Different aspect are presented vaguely and it is hard to make a clear link with the current study.
We added the the following sentences :
Lueth et al. identified 34 mtDNA somatic mutations in 84% (16/19) patients with pilocytic astrocytoma. 17 from this 34 mutation were found in genes of oxidative phosphorylation. Three of them were missense mutations ( CO1 (L112 M), cytochrome b (L236I) , ATP6 (M60 V) [11].
Results
- The results should be organized more clearly. They are presented more like a list of results. The paper includes too many tables, with list of results, which should be part of supplementary data –
- It is very difficult to follow the idea with the results presented this way.
It was corrected – The number of tables has been reduced to 11
Supplementary Materials: Figure 1. Reference protein CO1 according to the Cambridge sequence (red font indicates the places where the amino acid shift took place). Figure 2. The protein CO1 with polymorphism STP514K (red font indicates the shift of amino acids in the protein).
Discussion
It is not clear what the study presented here brings in addition of the results presented in Rhiannon et al. (2015) and other discussed here. –
The results of Rhiannon et al (2015) and Wallace et al. concern changes in mt DNA in brain tumors. The results are consistent with our results, indicating the role of the changes detected in the neoplastic process. We believe that for this reason this works should be presented in the discussion.
The following sentence was added: The similarity of the obtained results with the works cited above indicates that the described changes may be related to neoplasms in brain tumors.
The discussion is way too long, difficult to follow, and does not clearly show what the current study brings to the understanding of the role of mitochondria in brain tumor.
We removed some part of discussion. A conclusion is missing to finish the paper. – the conclusion was added to the end of discussion
Methods
The table 1 is cited at the end of the paper. The figures should be numbered in order of citation. –It was corrected-table 1 is 11 .

Reviewer 2 Report
The manuscript “Changes in mitochondrial DNA in genes of the respiratory complex III, IV and V in brain tumors in human” by Kozakiewicz et al. investigates the possible relationship between the occurrence of specific mtDNA polymorphisms in genes coding proteins of respiratory complexes III, IV and V and the presence of gliomas. Authors performed next-generation sequencing of mtDNA in samples of 30 patients diagnosed with grade II, III or IV glioma and in silico analysis of the polymorphisms they found. Using this methodology, they propose that several changes within MT-CYB and MT-CO1 genes could affect the carcinogenesis process. Besides, authors propose a polymorphism in MT-ATP6 gene as potentially harmful.
The analysis was well conducted, and conclusions are supported by results. However, I have detected some mistakes that make the work difficult to understand, beginning for the article title. Besides, English spelling and grammar should be revised.
Comments:
- The article’s title does not reflect its content. It should indicate that changes in the sequence of mt-DNA encoded subunits of respiratory complexes III, IV and V could be related with brain tumours in humans.
- The abstract is also difficult to understand. For example, it says ATP carrier (line 17) instead of ATP or “of specific mitochondrial DNA polymorphisms of genes III, IV and V of the respiratory complex and gliomas of the brain in the II, III and IV grade according to WHO” (line 23) instead of “…. polymorphisms of genes coding of complexes III, IV and V subunits”.
The aim of the study should be indicated before the description of materials or techniques.
- The introduction also contains several mistakes such as:
o Line 44: “… for oxidation of hydroxyquinones Qo”.
o Line 64: “final component of mitochondrial electron transport” should be “…. transport chain”
o Line 72: It is not correct talking about the activity of mitochondrial cytochrome oxidase subunits. It should be the activity of the whole complex.
- In the tables there are some words written in Polish and the word "missense" is written missens in all of them.
Author Response
We would like to thank the reviewers for their generous comments on the manuscript. We have edited the manuscript to address their concerns. Point-by-point responses to reviewers’ comments are listed below this letter
Reviewer 2
The article’s title:
The article’s title does not reflect its content. It should indicate that changes in the sequence of mt-DNA encoded subunits of respiratory complexes III, IV and V could be related with brain tumours in humans.
The title was changed to: Mitochondrial DNA changes in genes of respiratory complexes III, IV and V could be related with brain tumours in humans.
The abstract
is also difficult to understand. For example, it says ATP carrier (line 17) instead of ATP or “of specific mitochondrial DNA polymorphisms of genes III, IV and V of the respiratory complex and gliomas of the brain in the II, III and IV grade according to WHO” (line 23) instead of “…. polymorphisms of genes coding of complexes III, IV and V subunits”. –
The abstract has been changed as follows
Changes in mitochondrial DNA can contribute to both increasing and decreasing the likelihood of cancer developing. This process is complex and not fully understood. The study aimed to identify the possible relationships between the occurrence of specific mitochondrial DNA polymorphisms and mutations in respiratory complexes III, IV and V in brain gliomas. The investigated material included blood and tumor material collected from 30 patients diagnosed with WHO grade II, III or IV glioma. Based on the collected material, the mitochondrial genetic variants were investigated using next-generation sequencing for the entire mitochondrial genome with the MiSeq / FGx system (Illumina). The study investigated the in silico effects of missense mutations on the biochemical properties, structure and functioning of the encoded protein, as well as their potential harmfulness. The A14793G (MT-CYB), A15758G, (MT-CYB), A15218G (MT-CYB), G7444A (MT-CO1) polymorphisms and the T15663C (MT-CYB) mutations were assessed as harmful in silico alterations that could be involved in carcinogenesis. The T15663C (MT-CYB) mutation, which belongs to the mitochondrial subgroups of haplogroups (M, S, L1) outside the Caucasian population, may be associated with a predisposition to brain tumors in this population. The G8959A (E145K) ATP6 missense mutation was assessed as potentially harmful.
The aim of the study should be indicated before the description of materials or techniques –It was corrected
The introduction:
Line 44: “… for oxidation of hydroxyquinones Qo”. – was corrected as for oxidation of hydroxyquinones (site Qo)
Line 64: “final component of mitochondrial electron transport” should be “…. transport chain” –- It was corrected
Line 72: It is not correct talking about the activity of mitochondrial cytochrome oxidase subunits. It should be the activity of the whole complex. – It was corrected
In the tables there are some words written in Polish and the word "missense" is written missens in all of them-It was corrected

Round 2
Reviewer 1 Report
None
Author Response
Response to Reviewer 1
We would like to thank the reviewer for their generous comments on the manuscript. As (Rewiever -1 second round) there were no comments or suggestions for the authors, we considered that the reviewer had maintained all his previous comments. The work has been re-submitted for linguistic proofreading - attached certificate. We improved the manuscript by adding some fragments and reduced the number of tables. We do not know what comments reviewer has on the method (are the methods adequately described? -Must be improved), apart from the mismatch of the table number, which has been corrected. We also cannot agree with the reviewer on the discussion
Response to reviewer’s comments are listed below this letter
Abstract:
was corrected- the following sentence was added:
Polymorphism and mutations, especially those of missense type can affect mitochondrial faunction, particulary, if the conservative domain of the protein is concerned.
Introduction was corrected and also the following sentences were added
Mitochondrial haplogroups were shown to play a role in neurological diseases. The risk of Parkinson’s disease is higher in people with haplogroups J and T characterised by T to C substitution at position 4216 of the ND1 gene [20],[21]. As regards, neurodegenerative conditions, one study involving a European population has shown an increase in the risk of Alzheimer’s disease among males from haplogroup U, compared to haplogroup H, the most common haplogroup in Europe [22].
The aim of this study was to identify the possible relationships between brain gliomas in humans and the occurrence of specific mitochondrial DNA polimorphisms and mutations in respiratory complexes III, IV and V and
Results- were corrected-The number of table was delated to 6
Methods- We do not know what criticisms the reviewer has about the method (are the methods adequately described?-must be improved), apart from the incompatibility of the table number, which has been corrected
Discussion- We removed some part of discussion.
The similarity of our results to the cited works suggests that some of the detected changes may be related to brain tumors. Changes in mtDNA that occur as mutations and polymorphisms in various cancers indicate their role in cancer development. Failure to present studies on other types of neoplasms would change the nature and purpose of the study, including demonstrating whether the observed changes may be related to brain gliomas. Demonstrating the effect of changes on the properties of a protein may also indicate a their relationship with the neoplastic process.

Round 3
Reviewer 1 Report
The paper is still very hard to read and not presenting the story clearly.
The abstract is better but end with a list of results without a general sentence.
The discussion is still too long and difficult to follow.
Author Response
Response to Reviewer 1
We thank the reviewer for generous comments on the manuscript. We have edited the manuscript in accordance with the reviewer's comments. The work has been re-submitted for linguistic proofreading. We improved the manuscript by adding and removing some fragments.
Response to reviewer’s comments are listed below this letter
1)Abstract:
The abstract is better but end with a list of results without a general sentence.
It was corrected and the general sentence was added :
In light of these results, further research into the role of mtDNA changes in brain tumors should be conducted
2)Results- were corrected- some sentences were changed.
3) Discussion
The discussion is still too long and difficult to follow.
We removed some part of discussion- 268 words were removed (from 2782 to 2514 words) and summary was changed as follows:
Three MT-CYB polymorphisms: A14793G, A15758G and A15218G, as well as the MT-CO1 G7444A MT-CYB T15663C and G8959A (ATP6) mutations may affect the function of mitochondria, thus potentially playing an important in the formation of brain tumors. The T15663C (MT-CYB) mutation may be a predisposing factor in brain tumours for the Caucasian population, since it belongs only to the mitochondrial subgroups of haplogroups (M, S, L1) outside this population.

Round 4
Reviewer 1 Report
None